# Directed Acyclic Graph Neural Networks

**Veronika Thost & Jie Chen**[*]
MIT-IBM Watson AI Lab, IBM Research
Veronika.Thost@ibm.com, chenjie@us.ibm.com

## Abstract

Graph-structured data ubiquitously appears in science and engineering. Graph neural networks (GNNs) are designed to exploit the relational inductive bias exhibited in graphs; they have been shown to outperform other forms of neural networks in scenarios where structure information supplements node features. The most common GNN architecture aggregates information from neighborhoods based on message passing. Its generality has made it broadly applicable. In this paper, we focus on a special, yet widely used, type of graphs—DAGs—and inject a stronger inductive bias—partial ordering—into the neural network design. We propose the *directed acyclic graph neural network*, DAGNN, an architecture that processes information according to the flow defined by the partial order. DAGNN can be considered a framework that entails earlier works as special cases (e.g., models for trees and models updating node representations recurrently), but we identify several crucial components that prior architectures lack. We perform comprehensive experiments, including ablation studies, on representative DAG datasets (i.e., source code, neural architectures, and probabilistic graphical models) and demonstrate the superiority of DAGNN over simpler DAG architectures as well as general graph architectures.

## 1 Introduction

Graph-structured data is ubiquitous across various disciplines (Gilmer et al., 2017; Zitnik et al., 2018; Sanchez-Gonzalez et al., 2020). Graph neural networks (GNNs) use both the graph structure and node features to produce a vectorial representation, which can be used for classification, regression (Hu et al., 2020), and graph decoding (Li et al., 2018; Zhang et al., 2019). Most popular GNNs update node representations through iterative message passing between neighboring nodes, followed by pooling (either flat or hierarchical (Lee et al., 2019; Ranjan et al., 2020)), to produce a graph representation (Li et al., 2016; Kipf & Welling, 2017; Gilmer et al., 2017; Veličković et al., 2018; Xu et al., 2019). The relational inductive bias (Santoro et al., 2017; Battaglia et al., 2018; Xu et al., 2020)—neighborhood aggregation—empowers GNNs to outperform graph-agnostic neural networks. To facilitate subsequent discussions, we formalize a message-passing neural network (MPNN) architecture, which computes representations $h_v^\ell$ for all nodes $v$ in a graph $\mathcal{G}$ in every layer $\ell$ and a final graph representation $h_\mathcal{G}$, as (Gilmer et al., 2017):

$$h_v^\ell = \text{COMBINE}^\ell\left(h_v^{\ell-1}, \text{AGGREGATE}^\ell\left(\underline{\{h_u^{\ell-1} \mid u \in \mathcal{N}(v)\}}\right)\right), \quad \ell = 1, \ldots, L, \quad (1)$$

$$h_\mathcal{G} = \text{READOUT}\left(\{h_v^L, v \in \mathcal{V}\}\right), \tag{2}$$

where $h_v^0$ is the input feature of $v$, $\mathcal{N}(v)$ denotes a neighborhood of node $v$ (sometimes including $v$ itself), $\mathcal{V}$ denotes the node set of $\mathcal{G}$, $L$ is the number of layers, and $\text{AGGREGATE}^\ell$, $\text{COMBINE}^\ell$, and $\text{READOUT}$ are parameterized neural networks. For notational simplicity, we omit edge attributes; but they can be straightforwardly incorporated into the framework (1)–(2).

Directed acyclic graphs (DAGs) are a special type of graphs, yet broadly seen across domains. Examples include parsing results of source code (Allamanis et al., 2018), logical formulas (Crouse et al., 2019), and natural language sentences, as well as probabilistic graphical models (Zhang et al., 2019), neural architectures (Zhang et al., 2019), and automated planning problems (Ma et al., 2020).

---

[*]To whom correspondence should be addressed.

A directed graph is a DAG if and only if the edges define a partial ordering over the nodes. The partial order is an additionally strong inductive bias one naturally desires to incorporate into the neural network. For example, a neural architecture seen as a DAG defines the acyclic dependency of computation, an important piece of information when comparing architectures and predicting their performance. Hence, this information should be incorporated into the architecture representation for higher predictive power.

In this work, we propose DAGNNs—directed acyclic graph neural networks—that produce a representation for a DAG driven by the partial order. In particular, the order allows for updating node representations based on those of *all* their predecessors sequentially, such that nodes without successors digest the information of the entire graph. Such a processing manner substantially differs from that of MPNNs where the information landed on a node is limited by a multi-hop local neighborhood and thus restricted by the depth $L$ of the network.

Modulo details to be elaborated in sections that follow, the DAGNN framework reads

$$h_v^\ell = F^\ell\Big(h_v^{\ell-1}, G^\ell\big(\underline{\{h_u^\ell \mid u \in \mathcal{P}(v)\}, h_v^{\ell-1}}\big)\Big), \quad \ell = 1, \dots, L, \tag{3}$$

$$h_{\mathcal{G}} = R\Big(\{h_v^\ell, \ell = 0, 1, \dots, L, v \in \mathcal{T}\}\Big), \tag{4}$$

where $\mathcal{P}(v)$ denotes the set of direct predecessors of $v$, $\mathcal{T}$ denotes the set of nodes without (direct) successors, and $G^\ell$, $F^\ell$, and $R$ are parameterized neural networks that play similar roles to AGGREGATE$^\ell$, COMBINE$^\ell$, and READOUT, respectively.

A notable difference between (3)–(4) and (1)–(2) is that the superscript $\ell - 1$ inside the underlined part of (1) is advanced to $\ell$ in the counterpart in (3). In other words, MPNN aggregates neighborhood information from the past layer, whereas DAGNN uses the information in the current layer. An advantage is that DAGNN always uses more recent information to update node representations.

Equations (3)–(4) outline several other subtle but important differences between DAGNN and MPNNs, such as the use of only direct predecessors for aggregation and the pooling on only nodes without successors. All these differences are unique to the special structure a DAG enjoys. Exploiting this structure properly should yield a more favorable vectorial representation of the graph. In Section 2, we will elaborate the specifics of (3)–(4). The technical details include (i) attention for node aggregation, (ii) multiple layers for expressivity, and (iii) topological batching for efficient implementation, all of which yield an instantiation of the DAGNN framework that is state of the art.

For theoretical contributions, we study topological batching and justify that this technique yields maximal parallel concurrency in processing DAGs. Furthermore, we show that the mapping defined by DAGNN is invariant to node permutation and injective under mild assumptions. This result reassures that the graph representation extracted by DAGNN is discriminative.

Because DAGs appear in many different fields, neural architectures for DAGs (including, notably, D-VAE (Zhang et al., 2019)) or special cases (e.g., trees) are scattered around the literature over the years. Generally, they are less explored compared to MPNNs; and some are rather application-specific. In Section 3, we unify several representative architectures as special cases of the framework (3)–(4). We compare the proposed architecture to them and point out the differences that lead to its superior performance.

In Section 4, we detail our comprehensive, empirical evaluation on datasets from three domains: (i) source code parsed to DAGs (Hu et al., 2020); (ii) neural architecture search (Zhang et al., 2019), where each architecture is a DAG; and (iii) score-based Bayesian network learning (Zhang et al., 2019). We show that DAGNN outperforms many representative DAG architectures and MPNNs.

Overall, this work contributes a specialized graph neural network, a theoretical study of its properties, an analysis of a topological batching technique for enhancing parallel concurrency, a framework interpretation that encompasses prior DAG architectures, and comprehensive evaluations. Supported code is available at `https://github.com/vthost/DAGNN`.

## 2 THE DAGNN MODEL

A *DAG* is a directed graph without cycles. Denote by $\mathcal{G} = (\mathcal{V}, \mathcal{E})$ a DAG, where $\mathcal{V}$ and $\mathcal{E} \subset \mathcal{V} \times \mathcal{V}$ are the node set and the edge set, respectively. A (strong) *partial order* over a set $S$ is a binary

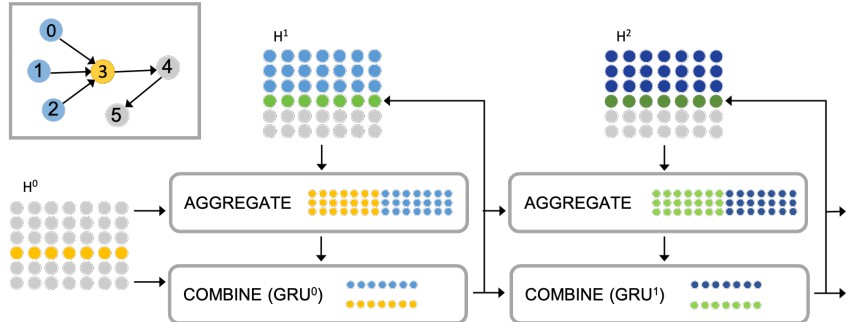

Figure 1: Processing of node $v = 3$ (orange). For each layer $\ell$, we collect representations $h_v^\ell$ for all nodes $v$ in a matrix $\mathcal{H}^\ell$, where each row represents one node. The initial feature matrix is $\mathcal{X} = \mathcal{H}^0$. In the first layer, the representations of the direct predecessors $\mathcal{P}(v) = \{0, 1, 2\}$ (blue) have been computed; they are aggregated together with the past representation of $v$ (orange) to produce a message. The GRU treats the message as the hidden state and the past representation of $v$ as input and outputs an updated representation for $v$ (green). This new representation will be used by $v$'s direct successors $\{4\}$ in the same layer and also as input to the next layer. Note that the figure illustrates the processing of only one node. In practice, a batch of nodes is processed; see Section 2.2.

relation $\leq$ that is transitive and asymmetric. Some authors use reflexivity versus irreflexivity to distinguish *weak partial order* over *strong partial order*. To unify concepts, we forbid self-loops (which otherwise are considered cycles) in the DAG and mean strong partial order throughout. A set $S$ with partial order $\leq$ is called a *poset* and denoted by a tuple $(S, \leq)$.

A DAG $(\mathcal{V}, \mathcal{E})$ and a poset $(S, \leq)$ are closely related. For any DAG, one can define a unique partial order $\leq$ on the node set $\mathcal{V}$, such that for all pairs of elements $u, v \in \mathcal{V}$, $u \leq v$ if and only if there is a directed path from $u$ to $v$. On the other hand, for any poset $(S, \leq)$, there exists (possibly more than) one DAG that uses $S$ as the node set and that admits a directed path from $u$ to $v$ whenever $u \leq v$.

In a DAG, all nodes without (direct) predecessors are called *sources* and we collect them in the set $\mathcal{S}$. Similarly, all nodes without (direct) successors are called *targets* and we collect them in the set $\mathcal{T}$. Additionally, we let $\mathcal{X} = \{h_v^0, \ v \in \mathcal{V}\}$ be the set of input node features.

## 2.1 MODEL

The main idea of DAGNN is to process nodes according to the partial order defined by the DAG. Using the language of MPNN, at every node $v$, we "aggregate" information from its neighbors and "combine" this aggregated information (the "message") with $v$'s information to update the representation of $v$. The main differences to MPNN are that (i) we use the current-layer, rather than the past-layer, information to compute the current-layer representation of $v$ and that (ii) we aggregate from the direct-predecessor set $\mathcal{P}(v)$ only, rather than the entire (or randomly sampled) neighborhood $\mathcal{N}(v)$. They lead to a straightforward difference in the final "readout" also. In the following, we propose an instantiation of Equations (3)–(4). See Figure 1 for an illustration of the architecture.

**One layer.** We use the attention mechanism to instantiate the *aggregate operator* $G^\ell$. For a node $v$ at the $\ell$-th layer, the output message $m_v^\ell$ computed by $G^\ell$ is a weighted combination of $h_u^\ell$ for all nodes $u \in \mathcal{P}(v)$ at the same layer $\ell$:

$$\underbrace{m_v^\ell}_{\text{message}} := G^\ell\Big(\{h_u^\ell \mid u \in \mathcal{P}(v)\}, h_v^{\ell-1}\Big) = \sum_{u \in \mathcal{P}(v)} \alpha_{vu}^\ell \Big(\underbrace{h_v^{\ell-1}}_{\text{query}}, \underbrace{h_u^\ell}_{\text{key}}\Big) \underbrace{h_u^\ell}_{\text{value}}. \tag{5}$$

The weighting coefficients $\alpha_{vu}^\ell$ follow the query-key design in usual attention mechanisms, whereby the representation of $v$ in the past layer, $h_v^{\ell-1}$, serves as the query. Specifically, we define

$$\alpha_{vu}^\ell\Big(h_v^{\ell-1}, h_u^\ell\Big) = \operatorname*{softmax}_{u \in \mathcal{P}(v)} \Big(w_1^{\ell\top} h_v^{\ell-1} + w_2^{\ell\top} h_u^\ell\Big), \tag{6}$$

where $w_1^\ell$ and $w_2^\ell$ are model parameters. We use the additive form, as opposed to the usual dot-product form,[1] since it involves fewer parameters. An additional advantage is that it is straightforward to incorporate edge attributes into the model, as will be discussed soon.

The *combine operator* $F^\ell$ combines the message $m_v^\ell$ with the previous representation of $v$, $h_v^{\ell-1}$, and produces an updated representation $h_v^\ell$. We employ a recurrent architecture, which is usually used for processing data in sequential order but similarly suits processing in partial order:

$$h_v^\ell = F^\ell\left(h_v^{\ell-1}, m_v^\ell\right) = \text{GRU}^\ell\Big( \underbrace{h_v^{\ell-1}}_{\text{input}}, \overbrace{\underbrace{m_v^\ell}_{\text{state}}}^{\text{message}} \Big), \tag{7}$$

where $h_v^{\ell-1}$, $m_v^\ell$, and $h_v^\ell$ are treated as the input, past state, and updated state/output of a GRU, respectively. This design differs from most MPNNs that use simple summation or concatenation to combine the representations. It further differs from GG-NN (Li et al., 2016) (which also employs a GRU), wherein the roles of the two arguments are switched. In GG-NN, the message is treated as the input and the node representation is treated as the state. In contrast, we start from node features and naturally use them as inputs. The message tracks the processed part of the graph and serves better the role of a hidden state, being recurrently updated.

By convention, we define $G^\ell(\emptyset, \cdot) = 0$ for the aggregator so that for nodes with an empty direct-predecessor set, the message (or, equivalently, the initial state of the GRU) is zero.

**Bidirectional processing.** Just like in sequence models where a sequence may be processed by either the natural order or the reversed order, we optionally invert the directions of the edges in $\mathcal{G}$ to create a *reverse DAG* $\widetilde{\mathcal{G}}$. We will use the tilde notation for all terms related to the reverse DAG. For example, the representation of node $v$ in $\widetilde{\mathcal{G}}$ at the $\ell$-th layer is denoted by $\widetilde{h}_v^\ell$.

**Readout.** After $L$ layers of (bidirectional) processing, we use the computed node representations to produce the graph representation. We follow a common practice—concatenate the representations across layers, perform a max-pooling across nodes, and apply a fully-connected layer to produce the output. Different from the usual practice, however, we pull across only the target nodes and concatenate the pooling results from the two directions. Recall that the target nodes contain information of the entire graph following the partial order. Mathematically, the readout $R$ produces

$$h_\mathcal{G} = \text{FC}\Big( \underset{v \in \mathcal{T}}{\text{Max-Pool}} \big( \overset{L}{\underset{\ell=0}{\|}} \, h_v^\ell \big) \; \| \; \underset{u \in \mathcal{S}}{\text{Max-Pool}} \big( \overset{L}{\underset{\ell=0}{\|}} \, \widetilde{h}_u^\ell \big) \Big). \tag{8}$$

Note that the target set $\widetilde{\mathcal{T}}$ of $\widetilde{\mathcal{G}}$ is the same as the source set $\mathcal{S}$ of $\mathcal{G}$. If the processing is unidirectional, the right pooling in (8) is dropped.

**Edge attributes.** The instantiation of the framework so far has not considered edge attributes. It is in fact simple to incorporate them. Let $\tau(u, v)$ be the type of an edge $(u, v)$ and let $y_\tau$ be a representation of edges of type $\tau$. We insert this information during message calculation in the aggregator. Specifically, we replace the attention weights $\alpha_{vu}^\ell$ defined in (6) by

$$\alpha_{vu}^\ell\left(h_v^{\ell-1}, h_u^\ell\right) = \underset{u \in \mathcal{P}(v)}{\text{softmax}} \left( {w_1^\ell}^\top h_v^{\ell-1} + {w_2^\ell}^\top h_u^\ell + {w_3^\ell}^\top y_{\tau(u,v)} \right). \tag{9}$$

In practice, we experiment with slightly fewer parameters by setting $w_3^\ell = w_1^\ell$ and find that the model performs equally well. The edge representations $y_\tau$ are trainable embeddings of the model. Alternatively, if input edge features are provided, $y_{\tau(u,v)}$ can be replaced by a neural network-transformed embedding for the edge $(u, v)$.

## 2.2 Topological Batching

A key difference to MPNN is that DAGNN processes nodes sequentially owing to the nature of the aggregator $G^\ell$, obeying the partial order. Thus, for computational efficiency, it is important to maximally exploit concurrency so as to better leverage parallel computing resources (e.g., GPUs). One

---

[1]The usual dot-product form reads $\alpha_{vu}^\ell(h_v^{\ell-1}, h_u^\ell) = \text{softmax}(\langle {W_1^\ell}^\top h_v^{\ell-1}, {W_2^\ell}^\top h_u^\ell \rangle)$. We find that in practice the dot-product form and the additive form perform rather similarly, but the former requires substantially more parameters. We are indebted to Hyoungjin Lim who pointed out that, however, in the additive form, the query term will be canceled out inside the softmax computation.

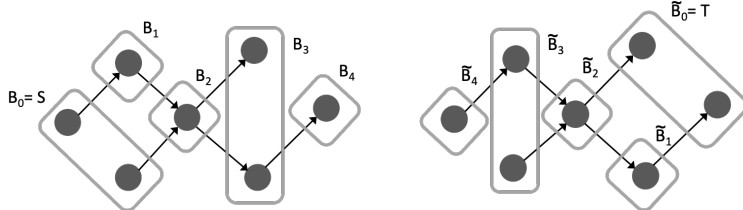

Figure 2: Topological batching. Left: for the original graph $\mathcal{G}$; right: for the reverse graph $\widetilde{\mathcal{G}}$.

observation is that nodes without dependency may be grouped together and processed concurrently, if their predecessors have all been processed. See Figure 2 for an illustration.

To materialize this idea, we consider *topological batching*, which partitions the node set $\mathcal{V}$ into ordered batches $\{\mathcal{B}_i\}_{i \geq 0}$ so that (i) the $\mathcal{B}_i$'s are disjoint and their union is $\mathcal{V}$; (ii) for every pair of nodes $u, v \in \mathcal{B}_i$ for some $i$, there is not a directed path from $u$ to $v$ or from $v$ to $u$; (iii) for every $i > 0$, there exists one node in $\mathcal{B}_i$ such that it is the tail of an edge whose head is in $\mathcal{B}_{i-1}$. The concept was proposed by Crouse et al. (2019);[2] in what follows, we derive several properties that legitimizes its use in our setting. First, topological batching produces the minimum number of sequential batches such that all nodes in each batch can be processed in parallel.

**Theorem 1.** *The number of batches from a partitioning that satisfies (i)–(iii) described in the preceding paragraph is equal to the number of nodes in the longest path of the DAG. As a consequence, this partitioning produces the minimum number of ordered batches such that for all $u \leq v$, if $u \in \mathcal{B}_i$ and $v \in \mathcal{B}_j$, then $i < j$. Note that the partial order $\leq$ is defined at the beginning of Section 2.*

The partitioning procedure may be as follows. All nodes without direct predecessors, $\mathcal{S}$, form the initial batch. Iteratively, remove the batch just formed from the graph, as well as the edges emitting from these nodes. The nodes without direct predecessors in the remaining graph form the next batch.

*Remark* 1. To satisfy Properties (i)–(iii), it is not necessary that $\mathcal{B}_0 = \mathcal{S}$; but the above procedure achieves so. Applying this procedure on the reverse DAG $\widetilde{\mathcal{G}}$, we obtain $\widetilde{\mathcal{B}}_0 = \mathcal{T}$. Note that the last batch for $\mathcal{G}$ may not be the same as $\mathcal{T}$; and the last batch for $\widetilde{\mathcal{G}}$ may not be the same as $\mathcal{S}$ either.

*Remark* 2. Topological batching can be straightforwardly extended to multiple graphs for better parallel concurrency: one merges the $\mathcal{B}_i$ for the same $i$ across graphs into a single batch. This is equivalent to treating the multiple DAGs as a single (albeit disconnected) DAG and applying topological batching on it.

### 2.3 PROPERTIES

In the following, we summarize properties of the DAGNN model; they are consistent with the corresponding results for MPNNs. To formalize these results, we let $\mathcal{M} : \mathcal{V} \times \mathcal{E} \times \mathcal{X} \to h_{\mathcal{G}}$ denote the mapping defined by Equations (3)–(4). For notational consistency, we omit bidirectional processing, and thus ignore the tilde term in (8). The first results state that DAGNN produces the same graph representation invariant to node permutation.

**Theorem 2.** *The graph representation $h_{\mathcal{G}}$ is invariant to node indexing if all $G^\ell$, $F^\ell$, and $R$ are so.*

**Corollary 3.** *The functions $G^\ell$, $F^\ell$, and $R$ defined in (5)–(8) are invariant to node indexing. Hence, the resulting graph representation $h_{\mathcal{G}}$ is, too.*

The next result states that the framework will not produce the same graph representation for different graphs (i.e., non-isomorphic graphs), under a common condition.

**Theorem 4.** *The mapping $\mathcal{M}$ is injective if $G^\ell$, $F^\ell$, and $R$, considered as multiset functions, are so.*

The condition required by Theorem 4 is not restrictive. There exist (infinitely many) injective multiset functions $G^\ell$, $F^\ell$, and $R$, although the ones instantiated by (5)–(8) are not necessarily injective. The modification to injection can be done by using the $\epsilon$-trick applied in GIN (Xu et al., 2019), but,

---

[2]See also an earlier implementation in `https://github.com/unbounce/pytorch-tree-lstm`

similar to the referenced work, the $\epsilon$ that ensures injection is unknown. In practice, it is either set as zero or treated as a tunable hyperparameter.

## 3 COMPARISON TO RELATED MODELS

In this section, we compare to the most closely related architectures for DAGs, including trees. Natural language processing is a major source of these architectures, since semantic parsing forms a rooted tree or a DAG. Recently, D-VAE (Zhang et al., 2019) has been suggested as a general-purpose autoencoder for DAGs. Its encoder architecture is the most similar one to ours, but we highlight notable differences that support the improvement DAGNN gains over the D-VAE encoder. All the models we compare with may be considered as restricted cases of the framework (3)–(4).

Rooted trees do usually not come with directed edges, because either direction (top-down or bottom-up) is sensible. Hence, we use the terminology "parent" and "child" instead. Unified under our framework, recursive neural networks tailored to trees (Socher et al., 2011; 2012; 2013; Ebrahimi & Dou, 2015) are applied to a fixed number of children when the aggregator acts on a concatenation of the child representations. Moreover, they assume that internal nodes do not come with input representations and hence the combine operator misses the first argument.

Tree-LSTM (Tai et al., 2015; Zhu et al., 2015; Zhang et al., 2016; Kiperwasser & Goldberg, 2016) and DAG-RNN (Shuai et al., 2016), like DAGNN, employ a recurrent architecture as the combine operator, but the message (hidden state) therein is a naive sum or element-wise product of child representations. In a variant of Tree-LSTM, the naive sum is replaced by a sum of child representations multiplied by separate weight matrices. A limitation of this variant is that the number of children must be the same and the children must be ordered. Another limitation is that both architectures assume that there is a single terminal node (in which case a readout is not invoked).

The most similar architecture to DAGNN is the encoder of D-VAE. There are two notable differences. First, D-VAE uses the gated sum as aggregator but we use attention which leverages the information of not only the summands ($h_u^\ell$) but also that of the node under consideration ($h_v^{\ell-1}$). This additional source of information enables attention driven by external factors and improves over self attention. Second, similar to all the aforementioned models, D-VAE does not come with a layer notion. On the contrary, we use multiple layers, which are more natural and powerful in the light of findings about general GNNs. Our empirical results described in the following section confirm so.

## 4 EVALUATION

In this section, we demonstrate the effectiveness of DAGNN on multiple datasets and tasks over a comprehensive list of baselines. We compare timing and show that the training cost of DAGNN is comparable with that of other DAG architectures. We also conduct ablation studies to verify the importance of its components, which prior DAG architectures lack.

### 4.1 DATASETS, TASKS, METRICS, AND BASELINES

The **OGBG-CODE** dataset (Hu et al., 2020) contains 452,741 Python functions parsed into DAGs. We consider the **TOK** task, predicting the tokens that form the function name; it is included in the Open Graph Benchmark (OGB). Additionally, we introduce the **LP** task, predicting the length of the longest path of the DAG. The metric for TOK is the F1 score and that for LP is accuracy. Because of the vast size, we also create a 15% training subset, **OGBG-CODE-15**, for similar experiments.

For this dataset, we consider three basic baselines and several GNN models for comparison. For the TOK task, the **Node2Token** baseline predicts tokens from the attributes of the second graph node, while the **TargetInGraph** baseline predicts tokens that appear in both the ground truth and in the attributes of some graph node. These baselines exploit the fact that the tokens form node attributes and that the second node's attribute contains the function name if it is part of the vocabulary. For the LP task, the **MajorityInValid** baseline constantly predicts the majority length seen from the validation set. The considered GNN models include four from OGB: **GCN** (Kipf & Welling, 2017), **GIN** (Xu et al., 2019), **GCN-VN**, **GIN-VN** (where -VN means adding a virtual node connecting all existing nodes); two using attention/gated-sum mechanisms: **GAT** (Veličković et al., 2018), **GG-NN**

Table 1: Prediction performance on the full dataset OGBG-CODE and a 15% subset OGBG-CODE-15 for two tasks: TOK and LP. **Best results are boldfaced** and second best are underlined.

| Model | TOK
F1 ↑ | TOK-15
F1 ↑ | LP
Acc ↑ | LP-15
Acc ↑ |
|---|---|---|---|---|
| Node2Token | $13.04_{\pm 0.00}$ | $13.04_{\pm 0.00}$ | - | - |
| TargetInGraph | $27.32_{\pm 0.00}$ | $27.08_{\pm 0.00}$ | - | - |
| MajorityInValid | - | - | $22.66_{\pm 0.00}$ | $22.66_{\pm 0.00}$ |
| GCN | $31.63_{\pm 0.18}$ | $24.39_{\pm 0.40}$ | $95.55_{\pm 0.62}$ | $90.66_{\pm 2.00}$ |
| GCN-VN | $32.63_{\pm 0.13}$ | $24.44_{\pm 0.25}$ | $96.62_{\pm 0.44}$ | $92.87_{\pm 1.19}$ |
| GIN | $31.63_{\pm 0.20}$ | $21.49_{\pm 0.61}$ | $98.36_{\pm 0.32}$ | $92.53_{\pm 2.30}$ |
| GIN-VN | $32.04_{\pm 0.18}$ | $21.10_{\pm 0.61}$ | $98.60_{\pm 0.23}$ | $93.27_{\pm 2.53}$ |
| GAT | $\underline{33.59}_{\pm 0.32}$ | $\underline{27.37}_{\pm 0.16}$ | $93.71_{\pm 0.24}$ | $83.15_{\pm 1.34}$ |
| GG-NN | $28.04_{\pm 0.27}$ | $23.15_{\pm 0.49}$ | $96.48_{\pm 0.27}$ | $89.16_{\pm 2.31}$ |
| SAGPool | $31.88_{\pm 0.39}$ | $24.45_{\pm 0.77}$ | $72.68_{\pm 14.29}$ | $60.66_{\pm 11.42}$ |
| ASAP | $28.30_{\pm 0.72}$ | $25.06_{\pm 0.37}$ | $87.84_{\pm 2.77}$ | $71.56_{\pm 3.76}$ |
| D-VAE | $32.64_{\pm 0.17}$ | $27.08_{\pm 0.39}$ | $\underline{99.90}_{\pm 0.02}$ | $\underline{99.78}_{\pm 0.01}$ |
| **DAGNN** | $\mathbf{34.41}_{\pm 0.38}$ | $\mathbf{29.11}_{\pm 0.44}$ | $\mathbf{99.93}_{\pm 0.01}$ | $\mathbf{99.86}_{\pm 0.04}$ |

Table 2: Predictive performance of latent representations for datasets NA and BN.

| Model | NA
RMSE ↓ | NA
Pearson's $r$ ↑ | BN
RMSE ↓ | BN
Pearson's $r$ ↑ |
|---|---|---|---|---|
| S-VAE | $0.521_{\pm 0.002}$ | $0.847_{\pm 0.001}$ | $0.499_{\pm 0.006}$ | $0.873_{\pm 0.002}$ |
| GraphRNN | $0.579_{\pm 0.002}$ | $0.807_{\pm 0.001}$ | $0.779_{\pm 0.007}$ | $0.634_{\pm 0.002}$ |
| GCN | $0.482_{\pm 0.003}$ | $0.871_{\pm 0.001}$ | $0.599_{\pm 0.006}$ | $0.809_{\pm 0.002}$ |
| DeepGMG | $0.478_{\pm 0.002}$ | $0.873_{\pm 0.001}$ | $0.843_{\pm 0.007}$ | $0.555_{\pm 0.003}$ |
| D-VAE | $\underline{0.375}_{\pm 0.003}$ | $\underline{0.924}_{\pm 0.001}$ | $\underline{0.281}_{\pm 0.004}$ | $\underline{0.964}_{\pm 0.001}$ |
| **DAGNN** | $\mathbf{0.264}_{\pm 0.004}$ | $\mathbf{0.964}_{\pm 0.001}$ | $\mathbf{0.122}_{\pm 0.004}$ | $\mathbf{0.993}_{\pm 0.000}$ |

(Li et al., 2016); two hierarchical pooling approaches using attention: **SAGPool** (Lee et al., 2019), **ASAP** (Ranjan et al., 2020); and the **D-VAE encoder**.

The **NA** dataset (Zhang et al., 2019) contains 19,020 neural architectures generated by the ENAS software. The task is to predict the architecture performance on CIFAR-10 under the weight-sharing scheme. Since it is a regression task, the metrics are RMSE and Pearson's $r$. To gauge performance with Zhang et al. (2019), we similarly train (unsupervised) autoencoders and use sparse Gaussian process regression on the latent representation to predict the architecture performance. DAGNN serves as the encoder and we pair it with an adaptation of the D-VAE decoder (see Appendix D). We compare to **D-VAE** and all the autoencoders compared therein: **S-VAE** (Bowman et al., 2016), **GraphRNN** (You et al., 2018), **GCN** (Zhang et al., 2019), and **DeepGMG** (Li et al., 2018).

The **BN** dataset (Zhang et al., 2019) contains 200,000 Bayesian networks generated by using the R package bnlearn. The task is to predict the BIC score that measures how well a BN fits the Asia dataset (Lauritzen & Spiegelhalter, 1988). We use the same metrics and baselines as for NA.

## 4.2 RESULTS AND DISCUSSION

**Prediction performance, token prediction (TOK), Table 1.** The general trend is the same across the full dataset and the 15% subset. DAGNN performs the best. GAT achieves the second best result, surprisingly outperforming D-VAE (the third best). Hence, using attention as aggregator during message passing benefits this task. On the 15% subset, only DAGNN, GAT, and D-VAE match or surpass the TargetInGraph baseline. Note that not all ground-truth tokens are in the vocabulary and thus the best achievable F1 is 90.99. Even so, all methods are far from reaching this ceiling performance. Furthermore, although most of the MPNN models (middle section of the table) use as many as five layers for message passing, the generally good performance of DAGNN and D-VAE indicates that DAG architectures not restricted by the network depth benefit from the inductive bias.

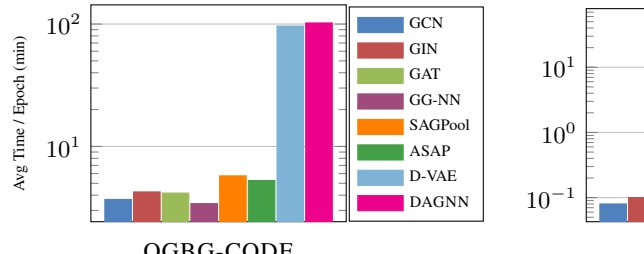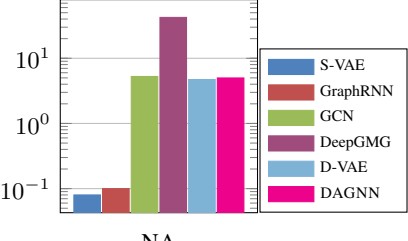

Figure 3: Average training time per epoch, on logarithmic scale. Standard deviation is negligible.

Table 3: Ablation results.

| Configuration | TOK-15 F1 ↑ | LP-15 Acc ↑ | NA RMSE ↓ | NA Pearson's $r$ ↑ | BN RMSE ↓ | BN Pearson's $r$ ↑ |
|---|---|---|---|---|---|---|
| **DAGNN** | **29.11**±**0.44** | 99.86±0.04 | **0.264**±**0.004** | **0.964**±**0.001** | 0.122±0.004 | 0.993±0.000 |
| Gated-sum aggr. | 24.98±0.45 | **99.88**±**0.02** | 0.451±0.002 | 0.887±0.001 | 0.486±0.005 | 0.878±0.001 |
| Single layer | 28.39±0.80 | 99.74±0.10 | 0.277±0.003 | 0.960±0.001 | 0.324±0.008 | 0.950±0.001 |
| FC layer | 26.08±0.80 | 99.85±0.02 | 0.280±0.004 | 0.959±0.001 | 0.362±0.002 | 0.934±0.001 |
| Pool all nodes | 28.40±0.08 | 99.78±0.05 | 0.302±0.002 | 0.952±0.001 | **0.098**±**0.003** | **0.996**±**0.001** |
| W/o edge attr. | 28.85±0.24 | 99.82±0.03 | - | - | - | - |

**Prediction performance, length of longest path (LP), Table 1.** This analytical task interestingly reveals that many of the findings for the TOK task do not directly carry over. DAGNN still performs the best, but the second place is achieved by D-VAE while GAT lags far behind. The unsatisfactory performance of GAT indicates that attention alone is insufficient for DAG representation learning. The hierarchical pooling methods also perform disappointingly, showing that ignoring nodes may modify important properties of the graph (in this case, the longest path). It is worth noting that DAGNN and D-VAE achieve nearly perfect accuracy. This result corroborates the theory of Xu et al. (2020), who state that when the inductive bias is aligned with the reasoning algorithm (in this case, path tracing), the model learns to reason more easily and achieves better sample efficiency.

**Prediction performance, scoring the DAG, Table 2.** On NA and BN, DAGNN also outperforms D-VAE, which in turn outperforms the other four baselines (among them, DeepGMG works the best on NA and S-VAE works the best on BN, consistent with the findings of Zhang et al. (2019).) While D-VAE demonstrates the benefit of incorporating the DAG bias, DAGNN proves the superiority of its architectural components, as will be further verified in the subsequent ablation study.

**Time cost, Figure 3.** The added expressivity of DAGNN comes with a tradeoff: the sequential processing of the topological batches requires more time than does the concurrent processing of all graph nodes, as in MPNNs. Figure 3 shows that such a trade-off is innate to DAG architectures, including the D-VAE encoder. Moreover, the figure shows that, when used as a component of a larger architecture (autoencoder), the overhead of DAGNN may not be essential. For example, in this particular experiment, DeepGMG (paired with the S-VAE encoder) takes an order of magnitude more time than does DAGNN (paired with the D-VAE decoder). Most importantly, not reflected in the figure is that DAGNN learns better and faster at larger learning rates, leading to fewer learning epochs. For example, DAGNN reaches the best performance at epoch 45, while D-VAE at around 200.

**Ablation study, Table 3.** While the D-VAE encoder performs competitively owing similarly to the incorporation of the DAG bias, what distinguishes our proposal are several architecture components that gain further performance improvement. In Table 3, we summarize results under the following cases: replacing attention in the aggregator by gated sum; reducing the multiple layers to one; replacing the GRUs by fully connected layers; modifying the readout by pooling over all nodes; and removing the edge attributes. One observes that replacing attention generally leads to the highest degradation in performance, while modifying other components yields losses too. There are two exceptions. One occurs on LP-15, where gated-sum aggregation surprisingly outperforms attention by a tight margin, considering the standard deviation. The other occurs on the modification of

Table 4: DAGNN results for different numbers of layers.

| | TOK-15 | LP-15 | NA | | BN | |
|---|---|---|---|---|---|---|
| # Layers | F1 ↑ | Acc ↑ | RMSE ↓ | Pearson's $r$ ↑ | RMSE ↓ | Pearson's $r$ ↑ |
| 1 | $28.39_{\pm 0.80}$ | $99.74_{\pm 0.10}$ | $0.277_{\pm 0.003}$ | $0.960_{\pm 0.001}$ | $0.324_{\pm 0.008}$ | $0.950_{\pm 0.001}$ |
| 2 | $\mathbf{29.11}_{\pm \mathbf{0.44}}$ | $\mathbf{99.86}_{\pm \mathbf{0.04}}$ | $0.264_{\pm 0.004}$ | $0.964_{\pm 0.001}$ | $\mathbf{0.122}_{\pm \mathbf{0.004}}$ | $\mathbf{0.993}_{\pm \mathbf{0.000}}$ |
| 3 | $28.96_{\pm 0.27}$ | $99.81_{\pm 0.06}$ | $\mathbf{0.260}_{\pm \mathbf{0.004}}$ | $\mathbf{0.965}_{\pm \mathbf{0.001}}$ | $0.129_{\pm 0.011}$ | $0.993_{\pm 0.001}$ |
| 4 | $28.91_{\pm 0.43}$ | $99.78_{\pm 0.04}$ | $0.265_{\pm 0.004}$ | $0.963_{\pm 0.001}$ | $0.129_{\pm 0.014}$ | $0.993_{\pm 0.002}$ |

the readout for the BN dataset. In this case, a Bayesian network factorizes the joint distribution of all variables (nodes) it includes. Even though the DAG structure characterizes the conditional independence of the variables, they play equal roles to the BIC score and thus it is possible that emphasis of the target nodes adversely affects the predictive performance. In this case, pooling over all nodes appears to correct the overemphasis.

**Sensitivity analysis, Table 4 and Figure 4.** It is well known that MPNNs often achieve best performance with a small number of layers, a curious behavior distinct from other neural networks. It is important to see if such a behavior extends to DAGNN. In Table 4, we list the results for up to four layers. One observes that indeed the best performance occurs at either two or three layers. In other words, one layer is insufficient (as already demonstrated in the ablation study) and more than three layers offer no advantage. We further extend the experimentation on TOK-15 with additional layers and plot the results in Figure 4. The trend corroborates that the most significant improvement occurs when going beyond a single layer. It is also interesting to see that a single layer yields the highest variance subject to randomization.

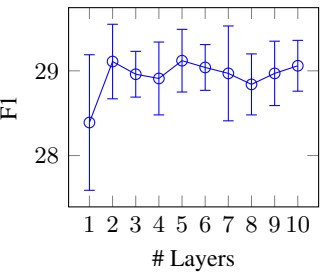

Figure 4: Extending Table 4 with further layers on TOK-15.

**Structure learning, Figure 5.** For an application of DAGNN, we extend the use of the BN dataset to learn the Bayesian network for the Asia data. In particular, we take the Bayesian optimization approach and optimize the BIC score over the latent space of DAGs. We use the graphs in BN as pivots and encode every graph by using DAGNN. The optimization yields a DAG with BIC score $-11107.29$ (see Figure 5). This DAG is almost the same as the ground truth (see Figure 2 of Lauritzen & Spiegelhalter (1988)), except that it does not include the edge from "visit to **A**sia?" to "**T**uberculosis?". It is interesting to note that the identified DAG has a higher BIC score than that of the ground truth, $-11109.74$. Furthermore, the BIC score is also much higher than that found by using the D-VAE encoder, $-11125.75$ (Zhang et al., 2019). This encouraging result corroborates the superior encoding quality of DAGNN and the effective use of it in downstream tasks.

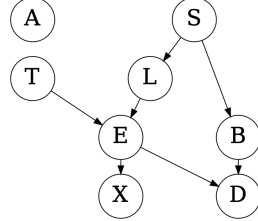

Figure 5: The Bayesian network identified by using Bayesian optimization over the latent space encoded by DAGNN.

## 5 CONCLUSIONS

We have developed DAGNN, a GNN model for a special yet widely used class of graphs—DAGs. It incorporates the partial ordering entailed by DAGs as a strong inductive bias towards representation learning. With the blessing of this inductive bias, we demonstrate that DAGNNs outperform MPNNs on several representative datasets and tasks. Through ablation studies, we also show that the DAGNN model is well designed, with several components serving as crucial contributors to the performance gain over other models that also incorporate the DAG bias, notably, D-VAE. Furthermore, we theoretically study a batching technique that yields maximal parallel concurrency in processing DAGs and prove that DAGNN is permutation invariant and injective.

ACKNOWLEDGMENTS

This work is supported in part by DOE Award DE-OE0000910. Most experiments were conducted on the Satori cluster (`satori.mit.edu`).

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

# A    PROOFS

*Proof of Theorem 1.* Let $(v_1, v_2, \ldots, v_d)$ be a longest path of the DAG. The number of batches must be at least $d$, because otherwise there exists a batch that contains at least two nodes on this path, violating Property (ii). On the other hand, given the partitioning, according to Property (iii), one may trace a directed path, one node from each batch, starting from the last one. The longest path must be at least that long. In other words, the number of batches must be at most the number of nodes on the longest path. Hence, these two numbers are equal. The consequence stated by the theorem straightforwardly follows.                                                                                                    □

*Proof of Theorem 2.* We first show that $h_v^\ell$ is invariant to the indexing of $v$ by double induction on $\ell$ and $v$. The base case is $\ell = 1$ and $v \in \mathcal{B}_0$. In this case, $m_v^1 = G^1(\emptyset, h_v^0) = 0$ is invariant to the indexing of $v$. Then, $h_v^1 = F^1(h_v^0, m_v^1)$ is, too. In the induction, if for all $\ell' < \ell$ and all $v'$, and for $\ell' = \ell$ and $v' \in \mathcal{B}_0 \cup \cdots \cup \mathcal{B}_{i-1}$, $h_{v'}^{\ell'}$ is invariant to the indexing of $v'$, then for $\ell' = \ell$ and $v \in \mathcal{B}_i$, $m_v^\ell = G^\ell(\{h_u^\ell \mid u \in \mathcal{P}(v)\}, h_v^{\ell-1})$ and $h_v^\ell = F^\ell(h_v^\ell, m_v^\ell)$ are both invariant to the indexing of $v$. Thus, by induction, for all $\ell' = \ell$ and all $v$, $h_v^{\ell'}$ is invariant to the indexing of $v$. Then, by an outer induction, for all $\ell$ and all $v$, $h_v^\ell$ is invariant to the indexing of $v$.

Therefore, $h_\mathcal{G} = R(\{h_v^\ell, \ell = 0, 1, \ldots, L, v \in \mathcal{T}\})$ is invariant to the indexing of the nodes in $\mathcal{T}$ and thus of the entire node set.                                                                                                    □

*Proof of Corollary 3.* The function $G^\ell$ is invariant to node indexing because it is a weighted sum of the elements in its first argument, $\{h_u^\ell\}$, whereas the weights are parameterized by using the same parameter $w_2^\ell$ for these elements.

The function $F^\ell$ is invariant to node indexing because its two arguments are clearly distinguished.

The function $R$ is invariant to node indexing because the FC layer applies to the pooling result of $h_v^\ell$ for a fixed set of $v$.                                                                                                    □

*Proof of Theorem 4.* Suppose two graphs $\mathcal{G}$ and $\mathcal{G}'$ have the same representation $h_\mathcal{G} = h_{\mathcal{G}'}$. Then, from the function $R$, they must have the same target set $\mathcal{T}$ and same node representations $h_v^\ell$ for all nodes $v \in \mathcal{T}$ and all layers $\ell$. In particular, for the last layer $\ell = L$, from the functions $F^L$ and $G^L$, each of these nodes, $v$, from the two graphs must have the same set of direct predecessors $\mathcal{P}(v)$, each element $u$ of which have the same representation $h_u^L$ across graphs. By backward induction, the two graphs must have the same node set $\mathcal{V}$ and edge set $\mathcal{E}$. Moreover, for each node $v \in \mathcal{V}$, the last-layer representation $h_v^L$ must be the same.

Furthermore, from the injection property of $F^\ell$, if a node $v$ shares the same node representation $h_v^\ell$ across graphs, then its past-layer representation $h_v^{\ell-1}$ must also be the same across graphs. A backward reduction traces back to the initial representation $h_v^0$, which concludes that the two graphs must have the same set of input node features $\mathcal{X}$.                                                                                                    □

# B    DATASET DETAILS

**OGBG-CODE.** The OGBG-CODE dataset was recently included in the Open Graph Benchmark (OGB) (Hu et al., 2020, Section 6.3). It contains 452,741 Python method definitions extracted from thousands of popular Github repositories. The method definitions are represented as DAGs by augmenting the abstract syntax trees with edges connecting the sequence of source code tokens. Hence, there are two types of edges. The min/avg/max numbers of nodes in the graphs are 11/125/36123, respectively. We use the node features provided by the dataset, including node type, attributes, depth in the AST, and pre-order traversal index.

The task suggested by Hu et al. (2020) is to predict the sub-tokens forming the method name, also known as "code summarization". The task is considered a proxy measure of how well a model captures the code semantics (Allamanis et al., 2018). We additionally consider the task of predicting the length of the longest path in the graph. We treat it as a 275-way classification because the maximum length is 275. The distribution of the lengths/classes is shown in Appendix E. To avoid triviality, for this task we remove the AST depth from the node feature set.

We adopt OGB's *project split*, whose training set consists of Github projects not seen in the validation and test sets. We also experiment with a subset of the data, OGBG-CODE-15, which contains only randomly chosen 15% of the OGBG-CODE training data. Validation and test sets remain the same.

In addition to OGBG-CODE, we further experiment with two DAG datasets, NA and BN, used by Zhang et al. (2019) for evaluating their model D-VAE. To compare with the results reported in the referenced work, we focus on the predictive performance of the latent representations of the DAGs obtained from autoencoders. We adopt the given 90/10 splits.

**Neural architectures (NA).** This dataset is created in the context of neural architecture search. It contains 19,020 neural architectures generated from the ENAS software (Pham et al., 2018). Each neural architecture has 6 layers (i.e., nodes) sampled from 6 different types of components, plus an input and output layer. The input node vectors are one-hot encodings of the component types. The weight-sharing accuracy (Pham et al., 2018) (a proxy of the true accuracy) on CIFAR-10 (Krizhevsky, 2009) is taken as performance measure. Details about the generation process can be found in Zhang et al. (2019, Appendix H).

**Bayesian networks (BN).** This dataset contains 200,000 random 8-node Bayesian networks generated by using the R package bnlearn (Scutari, 2010). The Bayesian Information Criterion (BIC) score is used to measure how well the DAG structure fits the Asia dataset (Lauritzen & Spiegelhalter, 1988). The input node vectors are one-hot encodings of the node indices according to topological sort. See Zhang et al. (2019, Appendix I) for further details.

## C  BASELINE DETAILS

**Baselines for OGBG-CODE.** We use three basic measures to set up baseline performance, two for token prediction and one for the longest path task. (1) **Node2Token**: This method uses the attribute of the second node of the graph as prediction. We observe that the second node either contains the function name, if the token occurs in the vocabulary (which is not always the case because some function names consist of multiple words), or contains "None". (2) **TargetInGraph**: This method pretends that it knows the ground-truth tokens but predicts only those occurring in the graph. One would expect that a learning model may be able to outperform this method if it learns the associations of tokens outside the current graph. (3) **MajorityInValid**: This method always predicts the majority length seen in the validation set.

Additionally, we compare with multiple GNN models. Some of them are the GNN implementations offered by OGB: **GCN**, **GIN**, **GCN-VN**, and **GIN-VN**. The latter two are extensions of the first two by including a *virtual node* (i.e., an additional node that is connected to all nodes in the graph). Note that the implementations do not strictly follow the architectures described in the original papers (Kipf & Welling, 2017; Xu et al., 2019). In particular, edge types are incorporated and inverse edges are added for bidirectional message passing.

Since our model features attention mechanisms, we include **GAT** (Veličković et al., 2018) and **GG-NN** (Li et al., 2016) for comparison. We also include two representative hierarchical pooling approaches, which use attention to determine node pooling: **SAGPool** (Lee et al., 2019) and **ASAP** (Ranjan et al., 2020). Lastly, we compare with the encoder of **D-VAE** (Zhang et al., 2019, Appendix E, F).

**Baselines for NA and BN.** Over NA and BN, we consider **D-VAE** and the baselines in Zhang et al. (2019, Appendix J). **S-VAE** (Bowman et al., 2016) applies a standard GRU-based RNN variational autoencoder to the topologically sorted node sequence, with node features augmented by the information of incoming edges, and decodes the graph by generating an adjacency matrix. **GraphRNN** (You et al., 2018) by itself serves as a decoder; we pair it with S-VAE encoder. **GCN** uses a GCN encoder while takes the decoder of D-VAE. **DeepGMG** (Li et al., 2018) similarly uses a GNN-based encoder but employs its own decoder (which is similar to the one in D-VAE). Note that all these baselines are autoencoders and our objective is to compare the performance of the latent representations.

## D    MODEL CONFIGURATIONS AND TRAINING

### D.1    EXPERIMENT PROTOCOL AND HYPERPARAMETER TUNING

Our evaluation protocols and procedures closely follow those of Hu et al. (2020); Zhang et al. (2019). For OGBG-CODE, we only changed the following. We used 5-fold cross validation due to the size of the dataset and the number of baselines for comparison. Since we compared with vast kinds of models in addition to the OGB baselines, we swept over a large range of learning rates and, for each model, picked the best from the set {1e-4, 5e-4, 1e-3, 15e-4, 2e-3, 5e-3, 1e-2, 15e-3} based on performance on OGBG-CODE-15. We stopped training when the validation metric did not improve further under a patience of 20 epochs, for all models but D-VAE and DAGNN. For the latter two, we used a patience of 10. Moreover, for these two models we used gradient clipping (at 0.25) due to the recurrent layers and a batch size of 80. Note that OGB uses 10-fold cross validation with a fixed learning rate of 1e-3, a fixed epoch number 30, and a batch size 128.

For NA and BN, we followed the exact training settings of Zhang et al. (2019, Appendix K). For DAGNN, we started the learning rate scheduler at 1e-3 (instead of 1e-4) and stopped at a maximum number of epochs, 100 for NA and 50 for BN (instead of 300 and 100, respectively). We also trained a sparse Gaussian process (SGP) (Snelson & Ghahramani, 2005) as the predictive model, as described in Zhang et al. (2019, Appendix L), to evaluate the performance of the latent representations. The prediction results were averaged over 10 folds.

For the Bayesian network learning experiment we similarly took over the settings of Zhang et al. (2019), running ten rounds of Bayesian optimization.

### D.2    BASELINE MODELS

All models were implemented in PyTorch (Paszke et al., 2019). For OGBG-CODE, we used the GCN and GIN models provided by the benchmark. We implemented a GAT model as described in Veličković et al. (2018) and GG-NN in Li et al. (2016). We used the SAGPool implementation of Lee et al. (2019) and ASAP from the Pytorch Geometric Benchmark Suite `https://github.com/rusty1s/pytorch_geometric/tree/master/benchmark`. All these models were implemented using PyTorch Geometric (Fey & Lenssen, 2019). We used the parameters suggested in OGB (e.g., 5 GNN layers, with embedding and hidden dimension 300), with the exception of ASAP where we used 3 instead of 5 layers due to memory constraints.

Since the D-VAE implementation does not support topological batching as we do, and also because of other miscellaneous restrictions (e.g., a single source node and target node), we reimplement D-VAE by using our DAGNN codebase. The reimplementation reproduces the results reported by Zhang et al. (2019). See Appendix F for more details.

### D.3    DAGNN IMPLEMENTATION

For DAGNN, we used hidden dimension 300. As suggested by OGB, we used independent linear classifiers to predict sub-tokens at each position of the sub-token sequence. Similarly, we used a linear classifier to predict the length of the longest path.

For the NA and BN datasets, we took the baseline implementations as well as training and evaluation procedures from Zhang et al. (2019). In particular, we used the corresponding configuration of D-VAE for the BN dataset. For DAGNN, we used the same hidden dimension 501 and adapted the decoder of D-VAE (by replacing the use of D-VAE encoder in part of the decoding process with our encoder). Additionally, we used bidirectional processing for token prediction over OGBG-CODE and for the experiment over BN. Since it did not offer improvement in performance for the longest path length prediction and for the experiment over NA but consumed too much time, for these cases we used unidirectional processing.

## E    DETAILS ON THE LONGEST PATH EXPERIMENT

We observe that for the MPNN baselines, the longest path results shown in Table 1 are much worse on the 15% subset than on the full dataset. We speculate whether the poorer performance is caused

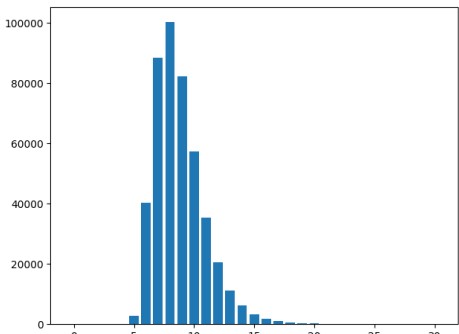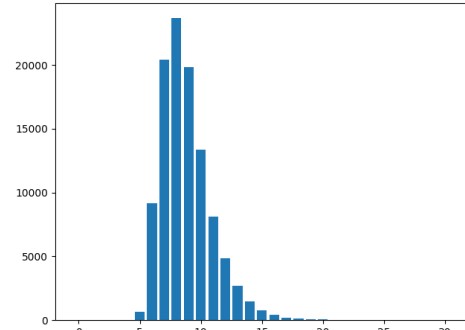

Figure 6: Distribution of the longest path lengths, for OGBG-CODE (left) and OGBG-CODE-15 (right). To improve readability, we ignored a tiny amount of graphs whose longest path length $> 30$. There are 58 such graphs in OGBG-CODE and 21 in OGBG-CODE-15.

by purely the size of training data, or additionally by the discrepancy of data distributions. Figure 6 shows that the data distributions are rather similar. Hence, we conclude that the degrading performance of MPNNs on a smaller training set is due to their low sample efficiency, in contrast to DAG architectures (D-VAE and DAGNN) that perform similarly on both the full set and the subset.

## F  REIMPLEMENTATION OF D-VAE

The original D-VAE implementation processes nodes sequentially and thus is time consuming. Therefore, we reimplement D-VAE by using our DAGNN codebase, in particular supporting topological batching. Table 5 shows that our reimplementation reproduces closely the results obtained by the original D-VAE implementation.

Table 5: Predictive performance of latent DAG representations for NA and BN. Comparison of the original implementation and our reimplementation.

|  | NA | | BN | |
|---|---|---|---|---|
| Model | RMSE | Pearson's $r$ | RMSE | Pearson's $r$ |
| D-VAE (orig) | $0.375_{\pm 0.003}$ | $0.924_{\pm 0.001}$ | $0.281_{\pm 0.004}$ | $0.964_{\pm 0.001}$ |
| D-VAE (ours) | $0.375_{\pm 0.004}$ | $0.925_{\pm 0.001}$ | $0.219_{\pm 0.003}$ | $0.977_{\pm 0.000}$ |

## G  ADDITIONAL ABLATION RESULTS

As mentioend in the main text, bidirectional processing is optional; it does not necessarily improve over unidirectional. Indeed, Table 6 shows that bidirectional works better on TOK-15 and BN, but unidirectional works better on LP-15 and NA. However, either way, DAGNN outperforms all baselines reported in Table 1 and 2, with only one exception: on LP-15, D-VAE performs worse than unidirectional but better than bidirectional.

Table 6: Bidirectional vs. unidirectional processing.

|  | TOK-15 | LP-15 | NA | | BN | |
|---|---|---|---|---|---|---|
| Bidirectional? | F1 ↑ | Acc ↑ | RMSE ↓ | Pearson's $r$ ↑ | RMSE ↓ | Pearson's $r$ ↑ |
| No | $28.44_{\pm 0.19}$ | $\mathbf{99.85}_{\pm 0.02}$ | $\mathbf{0.264}_{\pm 0.004}$ | $\mathbf{0.964}_{\pm 0.001}$ | $0.146_{\pm 0.035}$ | $0.992_{\pm 0.001}$ |
| Yes | $\mathbf{29.11}_{\pm 0.44}$ | $99.50_{\pm 0.22}$ | $0.324_{\pm 0.003}$ | $0.945_{\pm 0.001}$ | $\mathbf{0.122}_{\pm 0.004}$ | $\mathbf{0.993}_{\pm 0.000}$ |

