# OpenReview forum: "Directed Acyclic Graph Neural Networks"
_ICLR.cc/2021/Conference — ICLR 2021 Poster_

### Official Review · AnonReviewer3 · 2020-10-21
**A good proposal defining GNNs for DAGs**

**Rating:** 7
**Confidence:** 3

**Review:**

The paper presents a graph neural network formulation specific for directed acyclic graphs. In this formulation, the aggregation function also considers the information about the current layer. The model considers nodes following a topological batching in order to have the information of the predecessors when calculating aggregation of a node.
The system has been compared with several competitors on three datasets. The results are good in terms of metrics and seem promising also in terms of training time.

The proposal is clearly defined, but I would be happy if the paper also contains a very simple example considering a small DAG and showing some steps of the computation.

The proofs seem to me correct and the datasets seem complex enough to make the tests significant. Graph Neural Networks have come to the fore in recent years as a promising approach to solve many tasks and this work opens a good line of research.

One possible cons is that it focuses only on directed graphs. But this can also be a pro. It is important to continue to test whether this choice will lead to a system that can achieve significantly better results than other systems that are more general and do not pose this limitation. To test this aspect, three datasets are not sufficient, but the results look promising.

As for the tests, unfortunately, I could not run the scripts. It seemed that I could not download/create the datasets.

I have only minor issues to highlight in the paper.
In formula (1), L is not defined, its definition is given later.
In figure 1 there are strange lines that should be removed.
On page 7, TargetInVocab baseline is considered but in the paper I can find only TargetInGraph. What is this baseline?

---

> ### Author Response · Authors · 2020-11-17
> **We thank the reviewer for their constructive comments! Please see our rebuttal below.**
>
> In what follows we respond to the questions point by point. We have updated the paper accordingly.
>
> > The proposal is clearly defined, but I would be happy if the paper also contains a very simple example considering a small DAG and showing some steps of the computation.
>
> We have augmented Figure 1 with the input DAG, which may offer more intuition about the computation steps.
>
> > As for the tests, unfortunately, I could not run the scripts. It seemed that I could not download/create the datasets.
>
> We are sorry for this. Our .gitignore file mistakenly excluded the NA and BN datasets when we were pushing them. The issue is fixed now. The OGB dataset is downloaded by a script at runtime; it does not occur in the repository. Please note that the download and preprocessing may take a while (~1h). If you have further questions, please do not hesitate to open a Github issue.
>
> > I have only minor issues to highlight in the paper. In formula (1), L is not defined, its definition is given later. In figure 1 there are strange lines that should be removed. On page 7, TargetInVocab baseline is considered but in the paper I can find only TargetInGraph. What is this baseline?
>
> All fixed. TargetInVocab is a typo of TargetInGraph.

---

> ### Comment · Area_Chair1 · 2020-11-23
> **Feedback necessary**
>
> Dear reviewer,
>
> The authors have responded to your comments below. Could you please go over the response and give feedback to the authors sometime soon? The interactive discussion deadline is this Tuesday and you will not be able to interact with the authors after the date.
>
> Thanks, AC

---

> > ### Comment · AnonReviewer3 · 2020-11-24
> > **Wrong reader**
> >
> > Sorry, I have already answered to the authors wrongly including only them as readers.
> >
> > However, they clarified all my doubt. My score does not change.

---

### Official Review · AnonReviewer2 · 2020-10-27
**Improved modifications to learn representations for DAGs**

**Rating:** 7
**Confidence:** 3

**Review:**

This work considers DAGNN for learning representations for DAGs. Compared with message-passing neural network, particularly D-VAE, there are three subtle and notable differences motivated by the properties of DAG: (i) attention for node aggregation, (ii) multiple layers for expressivity, and (iii) topological batching for efficient training. Some theoretic guarantees are established (similar to those in D-VAE). Experiments show an improved performance, and ablation studies validate the proposed modifications. I recommend a weak acceptance based on the present version and am happy to improve my rating if authors address my concerns.

1. For Eq. (3), authors claimed that 'An advantage is that DAGNN always uses more recent information to update node representations.' However, compared with the other modifications, I didn't see any place to validate this statement. Can the authors explain more here to validate it is indeed an advantage?

2. Theorem 2 and Corollary 3 are highly related and are suggested to be combined as one. Indeed, I did not see why 'Corollary 3' is a corollary.

3. 'Remark 2. Topological batching can be straightforwardly extended to multiple graphs for better parallel concurrency: one merges the Bi for the same i across graphs into a single batch.' Can author give more details here, when graphs have different lengths?

4. Authors proposed a parallel strategy for an efficient training. However, compared with D-VAE which is a sequential execution, the training time seems to be very close. Why? and what if DAGNN does not use the parallelization?

5. about the BN experiment: compared with RMSE, BIC is a more reasonable metric to evaluate the final performance of the obtained DAG, as a better RMSE may indicate spurious edges wrt. the true graph. Also, the authors use the BIC score as the criterion when finding an optimal DAG. So what are the BICs for the respective methods? How does it compare with the BIC of the true graph?

6. Also in the BN experiment part, 'Even though the DAG structure characterizes the conditional independence of the variables, they play equal roles to the BIC score and thus it is possible that emphasis of the target nodes adversely affects the predictive performance. In this case, pooling over all nodes appears to correct the overemphasis.' What does it mean by 'target nodes'? Can you also give more details about this reasoning and if possible, can you try other BN problems to validate this reasoning?

7. the overall writing is OK, but may be improved in several parts. For example, 1) the introduction part has many comparisons with other approaches when stating contributions; separating them into two parts may be more clear.  3) in appendix D, please use subsections for 'MODEL CONFIGURATIONS AND TRAINING' for an improved readability.


*** after reading rebuttal *** All my concerns are addressed, and I decide to improve my evaluation.

---

> ### Author Response · Authors · 2020-11-17
> **We thank the reviewer for their considerate comments! Please see our rebuttal below.**
>
> In what follows we respond to the questions point by point. We have updated the paper accordingly.
>
> > For Eq. (3), authors claimed that 'An advantage is that DAGNN always uses more recent information to update node representations.' However, compared with the other modifications, I didn't see any place to validate this statement. Can the authors explain more here to validate it is indeed an advantage?
>
> Using the updated node representations for aggregation naturally follows the partial ordering entailed by the DAG, the main inductive bias we model. See also the reply to the first question of AnonReviewer4. Modifying back to the use of past-layer representation for aggregation leads to an architecture similar in nature to GG-NN, which we have compared in Table 1. The performance of GG-NN is far less competitive.
>
> > Theorem 2 and Corollary 3 are highly related and are suggested to be combined as one. Indeed, I did not see why 'Corollary 3' is a corollary.
>
> Corollary 3 essentially confirms that the assumption of Theorem 2 holds. Theorem 2 treats $G^{\ell}$, $F^{\ell}$, and $R$ in (3)--(4) generically, while Corollary 3 concludes a result for $G^{\ell}$, $F^{\ell}$, and $R$ specifically defined in (5)--(8).
>
> > 'Remark 2. Topological batching can be straightforwardly extended to multiple graphs for better parallel concurrency: one merges the $B_i$ for the same $i$ across graphs into a single batch.' Can author give more details here, when graphs have different lengths?
>
> For example, consider a DAG with edges {(1,3), (2,3), (3,4), (3,5)} and another DAG with edges {(6,8}, (7,8)}. The first DAG admits topological batches {1,2}, {3}, and {4,5} while the second DAG {6,7} and {8}. Then, the merging results in batches {1,2,6,7}, {3,8}, and {4,5}. We essentially treat the two DAGs as a single DAG (albeit disconnected), and then apply topological batching on this DAG.
>
> > Authors proposed a parallel strategy for an efficient training. However, compared with D-VAE which is a sequential execution, the training time seems to be very close. Why? and what if DAGNN does not use the parallelization?
>
> As described in Appendix D, we use the original D-VAE implementation only over the NA and BN datasets, but the D-VAE code we ran over OGBG-CODE is our reimplementation with topological batching.
> This code runs faster than the original implementation because of higher parallel concurrency. Apart from a fairer comparison of accuracy, a side reason of the reimplementation is that the original implementation makes several assumptions on the input data, which do not hold for the OGBG-CODE dataset, although they may not be hard to fix. We validated that our reimplementation reproduced the results reported by the original paper (see Appendix F). Equipped with topological batching, the time for D-VAE is rather similar to that of DAGNN (see Figure 3, left).
>
> > about the BN experiment: compared with RMSE, BIC is a more reasonable metric to evaluate the final performance of the obtained DAG, as a better RMSE may indicate spurious edges wrt. the true graph. Also, the authors use the BIC score as the criterion when finding an optimal DAG. So what are the BICs for the respective methods? How does it compare with the BIC of the true graph?
>
> The RMSE metric we use here indicates the difference between our prediction of the BIC and the ground truth BIC. This experiment does not aim at finding the optimal DAG because it lacks the Bayesian optimization part. However, the prediction of BIC may be used to replace the BIC calculation when evaluating the response surface in Bayesian optimization.
>
> > Also in the BN experiment part, 'Even though the DAG structure ....' What does it mean by 'target nodes'? Can you also give more details about this reasoning and if possible, can you try other BN problems to validate this reasoning?
>
> The target nodes, as defined in the last paragraph of Section 2.1, are nodes without successors. The pooling is done on only these nodes in DAGNN. We attempted to explain why this pooling is enough for all datasets but BN. A plausible reason is that putting too much emphasis on these nodes may not be suitable for predicting the BIC score, because the implied factorization of the joint distribution involves every node “equally”. Validating this explanation may need several new BN datasets and perform extensive experimentation with them. Such experiments are beyond the scope of the paper, we feel.
>
> > the overall writing is OK, but may be improved in several parts. For example, 1) the introduction part has many comparisons with other approaches when stating contributions; separating them into two parts may be more clear. 3) in appendix D, please use subsections for 'MODEL CONFIGURATIONS AND TRAINING' for an improved readability.
>
> We have updated the paper accordingly.

---

> > ### Comment · AnonReviewer2 · 2020-11-23
> > **Confusion about BN experiments**
> >
> > Thanks for a detailed response and I'm mostly satisfied.
> >
> > For the BN experiments, the goal of D-VAE is to 'train a VAE model on the training Bayesian networks, and search in the latent space for Bayesian networks with high BIC scores using Bayesian optimization'. For the present work, the goal is different and is to 'predict the BIC score that measures how well a BN fits the Asia dataset (Lauritzen & Spiegelhalter, 1988). We use the same metrics and baselines as for NA.'.
> >
> > My understanding is to predict the BIC score for a particular DAG based on the DAG's embedding, right? However, the BIC score, assuming linear data models, can be calculated easily and fast, with high accuracy. My question is,  what is the use of predicting such scores if the subsequent structure learning task is not performed? Can the authors report the best BIC score and its DAG, using the proposed structure?

---

> > > ### Author Response · Authors · 2020-11-24
> > > **Additional Results: Structure Learning**
> > >
> > > Thank you for the feedback.
> > >
> > > Indeed, the aim here is to show that DAGNN helps predicting the BIC score better for a DAG based on its embedding. It is true that the BIC score can be calculated easily in some cases. Yet, BIC is only the regression target (metric) we chose for the BN data because it is easy to obtain. The goal is that DAGNN creates latent representations that reflect other, arbitrary regression targets equally well, especially ones that are harder to obtain
> > >
> > > Representation learning is an indispensable component of the Bayesian-optimization approach for optimizing DAGs, because numerical optimization in action is done in the Euclidean space rather than in the DAG space. And our experiments show that the embedding computed by DAGNN has a better quality than those computed by the compared methods.
> > >
> > > From a practical viewpoint, structure learning is indeed the next step in Bayesian network learning. Since our focus is on the encoding capability of DAGNN and not on graph decoding, we thought further experiments would not provide additional insights about DAGNN but rather distract the reader, because of additional factors such as the decoder and optimization methods used.
> > >
> > > Nevertheless, we ran Bayesian optimization experiments, and the results are encouraging: the best DAG found using the DAGNN encoding has a BIC of -11107.29. The resulting graph is very close to the ground truth as depicted at https://www.bnlearn.com/bnrepository/discrete-small.html#asia, it is only missing the edge from “asia” to “tub”. It is interesting to note that such a DAG has a higher BIC score than the one of the ground truth.

---

> > > > ### Comment · AnonReviewer2 · 2020-11-24
> > > > **Thanks for a quick reply**
> > > >
> > > > Thanks for a quick reply. All my concerns are addressed and I am happy to improve my rating.
> > > >
> > > > Please include the structure learning part in your next version which I believe can enrich the current paper.

---

> > > > > ### Author Response · Authors · 2020-11-25
> > > > > **Paper Updated**
> > > > >
> > > > > Thank you so much, we have updated the paper.

---

### Official Review · AnonReviewer4 · 2020-10-28
**Interesting work, needs some clarification**

**Rating:** 6
**Confidence:** 4

**Review:**

Summary:
This paper introduces a model, Directed Acyclic Graph Neural Network (DAGNN), which processes information according to the flow defined by partial order. DAGNN can be regarded as a special case of previous GNN models, but specific to directed acyclic graph structures. The authors prove that the model satisfies the properties desired by DAG-based graph representation learning.Then they study topology batching on the proposed model to maximize parallel concurrency in processing DAGs. A comprehensive empirical evaluation is conducted on datasets from three domains to verify its effectiveness.


Reasons for score: Given the ubiquity of directed acyclic graphs, DAG-based graph neural networks have potential impacts on various fields. The authors propose an elegant and effective deep learning framework to learn node and graph representations on DAGs. My major concerns are the clarity of the model design, additional ablation tests, and sensitivity studies (see cons below).

Pros:
1) The problem is interesting. Directed acyclic graphs are very common in the real world. An efficient and powerful DAG-based graph neural network is expected to solve many unsolved problems in various domains.

2) The paper is well written. The authors introduce the framework based on the existing message passing neural network, which makes it easy to follow. The authors also present how this work handles special characteristics of DAGs. The techniques of the model are clearly stated. In particular, the comparison with highly relevant previous work is well explained.

3) The authors provide sufficient experimental results, including comparative studies on four datasets with the state-of-the-art benchmarks and ablation tests, which show the effectiveness of the proposed framework.

4) This paper provides a theoretical analysis of properties of the proposed model, which demonstrates that graph representations extracted by the proposed model are  discriminative.

Cons:
1) The framework needs more explanation. The authors introduce a recurrent neural network (Eq. 7) for updating node representation layer by layer. The input and past states are defined by the node representation at the last layer and the aggregated information from its predecessor nodes (i.e., message). As the authors introduced, these two arguments are switched in existing work. It would be better to provide more details about the design.
-  One question is, how will the roles of two arguments affect the performance of the model?

2) Although the paper provides several ablation studies, I still suggest the authors to consider the following ablation studies to enhance the quality of the paper:
	- What is the performance of the proposed model by changing the type of attention mechanism? Will it be relatively stable when using dot product attention which involves fewer parameters?
	- If we change the recurrent architecture to a feedforward neural network, what is the performance of the proposed model? Recurrent models are usually relatively slow in the training process and sometimes unnecessary.
	- How much does Bidirectional processing in recurrent neural networks help in improving the performance?
If the author can explain the above problems, it will be beneficial for people to understand this model.

3) No sensitivity analysis. The authors provide detailed model configurations, yet it is expected to see hyperparameter tuning results. For example, the proposed framework is a multi-layer graph neural network. It would be nice to test the sensitivity of the number of graph layers. Such knowledge of message-passing neural networks may not apply to the DAG-based framework. It would be better to provide some sensitivity studies or theoretical proof.

4) A minor concern about the ablation study: why didn’t the authors report the results on the LP dataset?

---

> ### Author Response · Authors · 2020-11-17
> **We thank the reviewer for their thoughtful comments! Please see our rebuttal below.**
>
> In what follows we respond to the questions point by point. We have updated the paper accordingly.
>
> > The framework needs more explanation. The authors introduce a recurrent neural network (Eq. 7) for updating node representation layer by layer. The input and past states are defined by the node representation at the last layer and the aggregated information from its predecessor nodes (i.e., message). As the authors introduced, these two arguments are switched in existing work. It would be better to provide more details about the design.
>
> The design follows a natural intuition, inspired by recurrent neural networks (broadly speaking). Consider a simple example of the DAG---a chain graph---and imagine applying a GRU on it. The chain graph admits a unique sequence order for the nodes. When applying GRU on this sequence, in every step, it takes in an input---node feature---and a past hidden state and uses the input to update the state. Extending this idea to general DAGs, we still use the node feature as input, but now the hidden state becomes a set of hidden states, one for each direct predecessor of the node. Our message represents the aggregated hidden states from these direct predecessors (Eq. 5), which we consider as the intuitive choice.
>
> > Although the paper provides several ablation studies, I still suggest the authors to consider the following ablation studies to enhance the quality of the paper:
>
> We had done several of the suggested experiments but, since most of the results are as expected, we did not add them to the original submission. However, some indeed strengthen the paper. We added those results in Section 4 and the remaining ones in Appendix G.
>
> > What is the performance of the proposed model by changing the type of attention mechanism? Will it be relatively stable when using dot product attention which involves fewer parameters?
>
> The usual dot-product attention in Transformers requires more parameters: compared with Eq. 6, the parameter vectors $w_1^{\ell}$ and $w_2^{\ell}$ are changed to parameter matrices and the sum is changed to dot product. The performance of using dot product is highly comparable. For example, for TOK-15 we obtain 28.28 +/- 0.15 (vs. 29.11 +/- 0.44) and for LP-15 99.79 +/- 0.04 (vs. 99.86 +/- 0.04).
>
> > If we change the recurrent architecture to a feedforward neural network, what is the performance of the proposed model? Recurrent models are usually relatively slow in the training process and sometimes unnecessary.
>
> The performance of using feedforward layers is not better than that of using GRUs. We have updated Table 3 with the results. It shows nicely that recurrent layers are as suitable for DAG encoding as they are for encoding sequences.
>
> > How much does Bidirectional processing in recurrent neural networks help in improving the performance? If the author can explain the above problems, it will be beneficial for people to understand this model.
>
> We consider bidirectional processing as optional, as explained in the main text and in Appendix D. It works better than unidirectional processing on some datasets but not on all. We have updated the paper with an additional Table 6 in Appendix G with the comparison. Either way, these results are still better than all baselines, with only one exception: on LP-15, D-VAE performs worse than the unidirectional but better than the bidirectional DAGNN.
>
> > No sensitivity analysis. The authors provide detailed model configurations, yet it is expected to see hyperparameter tuning results. For example, the proposed framework is a multi-layer graph neural network. It would be nice to test the sensitivity of the number of graph layers. Such knowledge of message-passing neural networks may not apply to the DAG-based framework. It would be better to provide some sensitivity studies or theoretical proof.
>
> We have updated the paper with these experimental results (see Table 4 and Figure 4).  Note a few findings. First, underlining our contribution: results on all datasets indicate that using more than one layer is beneficial. Second, however, going deeper than two or three does not help. These observations are fairly consistent with those for MPNNs in general. Third, in our original results (Tables 1 and 2), we reported performance on two layers only. The new results suggest that better performance is obtained with three layers on the NA dataset.
>
> > A minor concern about the ablation study: why didn’t the authors report the results on the LP dataset?
>
> We have updated the paper with LP-15 results in Table 3. For this dataset, the use of gated-sum aggregation yields a marginally better result, with other ablated versions being less competitive as expected. Overall, across datasets, the proposed DAGNN components perform the best, although occasionally some ablated version works better for a particular dataset/task.

---

> ### Comment · Area_Chair1 · 2020-11-23
> **Feedback necessary**
>
> Dear reviewer,
>
> The authors have responded to your comments below. Could you please go over the response and give feedback to the authors sometime soon? The interactive discussion deadline is this Tuesday and you will not be able to interact with the authors after the date.
>
> Thanks, AC

---

### Comment · Area_Chair1 · 2020-11-23
**The end of the interactive discussion phase approaching**

Dear Reviewers,

The authors have provided detailed responses to your comments. Could you please go over the responses from the reviewers and provide feedback since the authors can have interactions with you only by this Tuesday (24th)?. I sincerely thank you for your service in reviewing for ICLR.

Thanks,
Area Chair

---

### Comment · ~Federico_Errica1 · 2021-05-25
**Relation to previous works**

Dear authors,

I have a genuine interest in the relationship between your work and the following well-known framework:

P. Frasconi, M. Gori and A. Sperduti, "A general framework for adaptive processing of data structures," in IEEE Transactions on Neural Networks, vol. 9, no. 5, pp. 768-786, Sept. 1998, doi: 10.1109/72.712151.

which was published shortly after this other one:

A. Sperduti and A. Starita, "Supervised neural networks for the classification of structures," in IEEE Transactions on Neural Networks, vol. 8, no. 3, pp. 714-735, May 1997, doi: 10.1109/72.572108.

Would you be so kind to help me understand the main differences between your work and these ones?

Best regards,
Federico Errica

---

> ### Comment · ~Jie_Chen1 · 2021-05-26
> **re:**
>
> This is a very open ended question. In a sense, the DAG structure defines a unique partial ordering, which restricts the "variety" of how information can flow from one end to the other. On the other hand, I personally like the concept of stacking layers. If you imagine the linearization of nodes based on the partial ordering as the x axis, stacking the layers gives you a y axis. Rather than flowing information from left to right, now you (sort of) flow information from lower left to upper right. It is this extra dimension of the flow makes the modeling more interesting.
>
> Another consideration is that for a node in the middle, it has a few ins and a few outs. How the incoming information is aggregated and how the outing information is spread are also interesting modeling questions. For in, we use some sort of attention; and for out, we just replicate the information along each outgoing edge. Philosophically, I like the concept of energy conservation. So if somehow the ins and the outs are balanced and nothing is leaked, it sounds more beautiful to me.
>
> Though, we need to be more considerate of what in and out means. The in consists of two parts: the actual input and the past system state. The out also consists of two parts: the updated system state and the output, although in RNNs these two things are often treated the same. The system state could not be conserved, so possibly f(input) + f(past state) = f(updated state) makes more sense, where f defines some space where energy conservation is obeyed.

---

> > ### Comment · ~Alessandro_Sperduti1 · 2021-06-02
> > **Body of work on Recursive Neural Networks for DAGs**
> >
> > Dear authors,
> >
> > your paper is completely missing a massive body of works on neural networks for DAGs (*partial* list provided below). Proper scientific work should not only refer recent papers, but also papers that have been published in the past, avoiding reinventing the wheel.
> >
> > A. Sperduti:
> > Encoding Labeled Graphs by Labeling RAAM. NIPS 1993: 1125-1132
> >
> > A. Sperduti, Darya Majidi, Antonina Starita:
> > Extended Cascade-Correlation for Syntactic and Structural Pattern Recognition. SSPR 1996: 90-99
> >
> > C. Goller, A. Küchler:
> > Learning task-dependent distributed representations by backpropagation through structure. ICNN 1996: 347-352
> >
> > A. Sperduti:
> > Neural Networks for Processing Data Structures. Summer School on Neural Networks 1997: 121-144
> >
> > P. Frasconi, M. Gori, A. Sperduti:
> > On the Efficient Classification of Data Structures by Neural Networks. IJCAI 1997: 1066-1071
> >
> > A. Sperduti:
> > An overview on supervised neural networks for structures. ICNN 1997: 2550-2554
> >
> > Enrico Francesconi, Paolo Frasconi, Marco Gori, Simone Marinai, Jianqing Sheng, Giovanni Soda, Alessandro Sperduti:
> > Logo Recognition by Recursive Neural Networks. GREC 1997: 104-117
> >
> > Alessandro Sperduti, Antonina Starita:
> > Supervised neural networks for the classification of structures. IEEE Trans. Neural Networks 8(3): 714-735 (1997)
> >
> > Alessandro Sperduti:
> > On the Computational Power of Recurrent Neural Networks for Structures. Neural Networks 10(3): 395-400 (1997)
> >
> > Paolo Frasconi, Marco Gori, Alessandro Sperduti:
> > A general framework for adaptive processing of data structures. IEEE Trans. Neural Networks 9(5): 768-786 (1998)
> >
> > Marco Gori, Andreas Küchler, Alessandro Sperduti:
> > On the implementation of frontier-to-root tree automata in recursive neural networks. IEEE Trans. Neural Networks 10(6): 1305-1314 (1999)
> >
> > Alessio Micheli, Diego Sona, Alessandro Sperduti:
> > Bi-Causal Recurrent Cascade Correlation. IJCNN (3) 2000: 3-8
> >
> > Marco Gori, Paolo Frasconi, Alessandro Sperduti:
> > Learning Efficiently with Neural Networks: A Theoretical Comparison between Structured and Flat Representations. ECAI 2000: 301-305
> >
> > Anna Maria Bianucci, Alessio Micheli, Alessandro Sperduti, Antonina Starita:
> > Application of Cascade Correlation Networks for Structures to Chemistry. Appl. Intell. 12(1-2): 115-145 (2000)
> >
> > Vincenzo Di Massa, Gabriele Monfardini, Lorenzo Sarti, Franco Scarselli, Marco Maggini, Marco Gori:
> > A Comparison between Recursive Neural Networks and Graph Neural Networks. IJCNN 2006: 778-785
> >
> > Markus Hagenbuchner, Ah Chung Tsoi, Alessandro Sperduti:
> > A Supervised Self-Organizing Map for Structured Data. WSOM 2001: 21-28
> >
> > Alessandro Sperduti:
> > Neural Networks for Adaptive Processing of Structured Data. ICANN 2001: 5-12
> >
> > Alessio Micheli, Alessandro Sperduti, Antonina Starita, Anna Maria Bianucci:
> > Analysis of the Internal Representations Developed by Neural Networks for Structures Applied to Quantitative Structure-Activity Relationship Studies of Benzodiazepines. J. Chem. Inf. Comput. Sci. 41(2): 202-218 (2001)
> >
> > Monica Bianchini, Marco Gori, Franco Scarselli:
> > Recursive Processing of Directed Acyclic Graphs. WIRN 2001: 96-101
> >
> > Alessandro Sperduti:
> > On Linear Separability of Sequences and Structures. ICANN 2002: 601-606
> >
> > Alessandro Vullo, Paolo Frasconi:
> > A Bi-Recursive Neural Network Architecture for the Prediction of Protein Coarse Contact Maps. CSB 2002: 187-196
> >
> > Alessio Micheli, Diego Sona, Alessandro Sperduti:
> > Formal Determination of Context in Contextual Recursive Cascade Correlation Networks. ICANN 2003: 173-180
> >
> > Markus Hagenbuchner, Alessandro Sperduti, Ah Chung Tsoi:
> > A self-organizing map for adaptive processing of structured data. IEEE Trans. Neural Networks 14(3): 491-505 (2003)
> >
> > A. Vullo, P. Frasconi:
> > A Recursive Connectionist Approach for Predicting Disulfide Connectivity in Proteins. SAC 2003: 67-71
> >
> > A. Micheli, D. Sona, A. Sperduti:
> > Contextual processing of structured data by recursive cascade correlation. IEEE Trans. Neural Networks 15(6): 1396-1410 (2004)
> >
> > Barbara Hammer, Alessio Micheli, Alessandro Sperduti, Marc Strickert:
> > A general framework for unsupervised processing of structured data. Neurocomputing 57: 3-35 (2004)
> >
> > Monica Bianchini, Marco Gori, Lorenzo Sarti, Franco Scarselli:
> > Recursive Neural Networks and Graphs: Dealing with Cycles. WIRN/NAIS 2005: 38-43
> >
> > Marco Gori, Alessandro Sperduti:
> > The loading problem for recursive neural networks. Neural Networks 18(8): 1064-1079 (2005)
> >
> > Barbara Hammer, Alessio Micheli, Alessandro Sperduti:
> > Universal Approximation Capability of Cascade Correlation for Structures. Neural Comput. 17(5): 1109-1159 (2005)
> >
> > Alessandro Sperduti:
> > Exact Solutions for Recursive Principal Components Analysis of Sequences and Trees. ICANN (1) 2006: 349-356
> >
> > Alessio Micheli, Alessandro Sperduti:
> > Recursive Principal Component Analysis of Graphs. ICANN (2) 2007: 826-835
> >
> > Alessandro Sperduti:
> > Efficient Computation of Recursive Principal Component Analysis for Structured Input. ECML 2007: 335-346

---

> > > ### Comment · ~Jie_Chen1 · 2021-06-03
> > > **re:**
> > >
> > > Thank you very much for pointing out the history and thank you for the enduring contribution to the subject. I agree in most of the times, if not all, scientific progress is built on past success. Coincidently, at this moment I am writing a paper and tracing a graph coarsening technique all the way back to 1939. And I am still not entirely certain if that was the first time the technique was introduced!
> > >
> > > On a broader note, the exploding number of papers in machine learning and AI gives me and us an unprecedented challenge on how to properly acknowledge prior work. I would estimate, for an average paper published in places like this, the number of closely related papers may reach hundreds, if not thousands. We may write a paper whose bibliography section is longer than the rest; but are there better solutions?
> > >
> > > I am thinking that we have citation graphs, we have google scholars, and we have all sorts of efforts to mine the literature. Would it be possible to build a system that can return a ranked list of papers for any given paper and tell people that, hey, the bibliography section is understandably incomplete; for more related work, check the list! For people who care about citations, we can even use the ranked list to define a new citation metric, and give authors whose work is less cited than justified, if any, a platform to shine.

---

### Decision · Program_Chairs · 2021-01-07
**Final Decision**

**Decision:**

Accept (Poster)

**Comment:**

This paper proposes a graph neural network architecture to learn representations for directed acyclic graphs. Specifically, the proposed method performs the aggregation of the representations from neighboring nodes in the topological order defined by the DAG, with a novel topological batching scheme, which allows to process the message passing operations in parallel. The authors propose theoretical analysis of the proposed methods, to show that it is invariant to node indexing and learns an injective mapping to discriminate between two different graphs. The proposed method is further experimentally validated on multiple tasks involving DAGs, and the results show that it outperforms existing GNNs, including existing methods that can capture DAGs such as D-VAE (encoder).

The reviewers were unanimously positive about the paper. All reviewers find the performance improvements and time-efficiency obtained with the proposed method to be satisfactory or promising, and one of the reviewers (R4) mentions that the tackled problem is important and the paper is well-written. However, there were concerns regarding insufficient explanations, missing ablation studies, and missing details of some parts of the proposed method. Yet, most of the issues have been satisfactorily addressed during the interactive discussion period. I agree with the reviewers that the paper is tackling an important problem, find the paper well-written, and the proposed DAGNN as practically useful. Thus I recommend an acceptance.

However, the contributions of the proposed work over D-VAE, which also deals with DAGs, should be better described, as also noted by R2. The DAGNN uses attention, and can stack multiple layers as it is a more general GNN framework while D-VAE is a generative model, but these seem like incremental differences over D-VAE, and it is not clear which contributes to DAGNN’s superior performance over D-VAE. Topological batching is a clear advantage of DAGNN over D-VAE, but the experimental results showing the advantage of it over D-VAE’s sequential training was missing in the original paper (while it was added later to the appendix). I suggest the authors to introduce D-VAE in the introduction, acknowledge that it also tackles DAGs, and clearly describe how the proposed method differs from D-VAE encoder in a separate section. Also, there needs to be an analysis on why the proposed DAGNN outperforms D-VAE, as well as time-efficiency comparison with the original D-VAE in the main text.

---

> ### Author Response · Authors · 2021-01-20
> **camera-ready version uploaded; highlights**
>
> We thank the program chairs for the support of this submission. In the final version, we have explicitly acknowledged D-VAE in the Introduction section. A detailed comparison is given in the Comparison section, alongside with the discussion of other models. We would like to stress the interpretation of this work as a framework (eqns. 3--4), which parallels and contrasts MPNN (eqns 1--2). We believe that the several enhancements over D-VAE, including the use of layers, attention, and topological batching, contribute to the better performance over D-VAE, as demonstrated in experiments and ablation studies.
>
> We are fond of the fact that the OGBG-CODE result tops the leaderboard (https://ogb.stanford.edu/docs/leader_graphprop/#ogbg-code) at the time this response is written. We are also fond of the fact that the work helps identify better Bayesian network structures, which come alongside with other attempts to address DAG structure learning problems by using GNNs (see e.g., http://proceedings.mlr.press/v97/yu19a.html)
>
> We hope that this work will inspire the community to advance GNNs over cases (e.g., DAGs) that appear frequently in practice. Further discussions and inquiries are welcome.